# The Clinical Potential of Oligonucleotide Therapeutics against Pancreatic Cancer

**DOI:** 10.3390/ijms20133331

**Published:** 2019-07-06

**Authors:** Kazuki Takakura, Atsushi Kawamura, Yuichi Torisu, Shigeo Koido, Naohisa Yahagi, Masayuki Saruta

**Affiliations:** 1Division of Gastroenterology and Hepatology, Department of Internal Medicine, The Jikei University School of Medicine, Tokyo 105-8461, Japan; 2Division of Research and Development for Minimally Invasive Treatment, Cancer Center, Keio University School of Medicine, Tokyo 160-8582, Japan

**Keywords:** oligonucleotide therapeutics, RNA interference, antisense, aptamer, decoy, pancreatic cancer

## Abstract

Although many diagnostic and therapeutic modalities for pancreatic cancer have been proposed, an urgent need for improved therapeutic strategies remains. Oligonucleotide therapeutics, such as those based on antisense RNAs, small interfering RNA (siRNA), microRNA (miRNA), aptamers, and decoys, are promising agents against pancreatic cancer, because they can identify a specific mRNA fragment of a given sequence or protein, and interfere with gene expression as molecular-targeted agents. Within the past 25 years, the diversity and feasibility of these drugs as diagnostic or therapeutic tools have dramatically increased. Several clinical and preclinical studies of oligonucleotides have been conducted for patients with pancreatic cancer. To support the discovery of effective diagnostic or therapeutic options using oligonucleotide-based strategies, in the absence of satisfactory therapies for long-term survival and the increasing trend of diseases, we summarize the current clinical trials of oligonucleotide therapeutics for pancreatic cancer patients, with underlying preclinical and scientific data, and focus on the possibility of oligonucleotides for targeting pancreatic cancer in clinical implications.

## 1. Introduction

“Oligonucleotide therapeutics” is a general term for state-of-the-art, molecular-target agents that employ chemically synthesized oligonucleotides with a single-stranded deoxyribonucleic acid (DNA) or ribonucleic acid (RNA) backbone with potential specificity. These agents can inhibit gene expression or impede protein function by binding to a specific sequence of a target gene or protein [1]. Therefore, oligonucleotide therapeutics have a high specificity and target molecules that cannot be controlled by conventional drugs, such as mRNA or noncoding RNA, resulting in the development of innovative drugs against cancers and genetic diseases. Representative oligonucleotide therapeutics include antisense oligonucleotides (ASOs), small interfering RNA (siRNA), microRNAs (miRNAs), aptamers, and decoys. Among them, ASOs and siRNAs have been developed further than the others.

The latest review articles show that pancreatic cancer is recognized as a malignant tumor with poor prognosis, due to delayed diagnosis and its high refractoriness to any available therapeutic approaches [2,3]. Thus, pancreatic cancer is receiving extensive attention in research on pharmaceutical agents, given the absence of satisfactory treatments for long-term survival and the increasing trend in diseases. Given this situation, the clinical applications of oligonucleotide therapeutics for patients with pancreatic cancer have been underway. Nine clinical trials with oligonucleotide therapeutics have been conducted for patients with pancreatic cancer with National Institutes of Health (NIH) approval; no trials with aptamers or decoys have been conducted. A clinical trial of miRNA-25 for the diagnosis of pancreatic cancer from China was approved by the NIH. A promising clinical trial with carbohydrate sulfotransferase 15 (CHST15)-siRNA (STNM01) for patients with pancreatic cancer was completed in Japan. In our previous study, CHST15, a specific enzyme that biosynthesizes chondroitin sulfate-E (CS-E), which is highly expressed in human pancreatic cancer cells, was found to be a key player in tumor growth, and the effects of STNM01 on tumor cell proliferation in vitro and growth in vivo were demonstrated [4].

Here, we summarize the current clinical trials of oligonucleotide therapeutics for patients with pancreatic cancer (Table 1), along with preclinical or scientific evidence of drug efficacy, focusing on the possibility of oligonucleotides working as a diagnostic or therapeutic agent for pancreatic cancer.

## 2. History of Oligonucleotide Therapeutics: From Pioneer Works to Current Diversity

Zamecnik et al. established the concept of ASOs [5,6]. Matsukura et al. promote the expansion of ASOs for various clinical applications [7]. Aptamers were first mentioned as a different kind of oligonucleotide in research about the regulation of human immunodeficiency virus-1 (HIV-1) in 1989 [8]. Thereafter, Fire et al. discovered RNA interference (RNAi) as a breakthrough biological process in *Caenorhabditis elegans* in 1998 [9]. Since then, RNAi has become a major tool in various types of laboratory studies; the first application of an RNAi strategy was successfully assessed using an siRNA directed against the mRNA encoding the N-protein of respiratory syncytial virus (RSV) [10]. miRNAs were found to be an option for oligonucleotides in 1993 [11], and the first clinical trial related to a decoy strategy has been performed for HIV-1 infection [12].

Taken together, studies from the past 20 years have produced diverse advancements in oligonucleotide research, and the potential targets for both diagnostic and therapeutic purposes might be unlimited based on the different mechanisms of action. Although the mechanisms by which oligonucleotides produce their effects mostly depend on their structure and chemistry (Table 2), the major premise of controlling the expression of targeted gene products is immutable. Due to the different mechanisms of each oligonucleotide, they act as inhibitors at different phases in pathogenesis. As shown in Figure 1, decoys work upstream of the expression process, targeting DNA coding transcription factors; subsequently, antisense DNA, siRNA, and miRNA act on the level of the target mRNA. Finally, aptamers directly inhibit the activity of proteins involved in pathogenesis. Their common point is that all of them act as inhibitors of pathogenesis by interfering with gene expression.

On the basis of preclinical research progress, the current clinical trials involving oligonucleotide therapeutics for patients with pancreatic cancer listed on the U.S. website http://Clinicaltrials.gov are summarized in Table 1. A greater level of detail about the different mechanisms of action of oligonucleotides is provided in the literature [13,14].

## 3. Antisense Oligonucleotides

When the antisense strategy was first introduced, ASOs, which are oligodeoxyribonucleotides, were recognized to represent a specific and systemic gene silencing strategy. Successful development of this strategy could allow an almost endless variety of human diseases to be treated, provided that a particular gene has been identified and characterized for the disease. Several advanced ASO and RNAi agents have already reached clinical trials; thus, we focus mainly on the clinical trials in this chapter.

ISIS-2503, a 20-mer ASO that hybridizes to the 5′-untranslated region of human *H-Ras* mRNA and inhibits *H-Ras* expression, was developed as a potential treatment for cancer, particularly cancers with abnormalities of *Ras* function [15]. Based on the evidence of the safety, potential effectiveness, and feasibility of ISIS-2503 administered intravenously for cancer patients in phase I trials, and of the rational approach of combining ISIS-2503 with gemcitabine, a conventional chemotherapy, to create a more effective regimen for pancreatic cancer [16,17], a multicenter phase II trial examining the combination of ISIS-2503 and gemcitabine in patients with metastatic or locally advanced pancreatic cancer was performed through the North Central Cancer Treatment Group [18]. The clinical study enrolled 48 pancreatic cancer patients, who showed a six-month survival percentage of 57.5%. The median survival was 6.6 months, and the response rate was 10.4%. The combined regimen was tolerable but insufficient.

The transforming growth factor-beta (TGF-β) signaling pathway is a key player in tumor progression [19], and TGF-β2 plays a pivotal role in the malignancy and progression of pancreatic cancer [20,21]. OT-101, an 18-mer phosphorothioate ASO, was designed for the targeted inhibition of human TGF-β2 mRNA, and OT-101 has suppressed TGF-β2 secretion in human pancreatic cancer cell lines [22,23,24]. Then, a phase I/II clinical trial of OT-101 administered intravenously in 37 patients with advanced pancreatic cancer, which showed a good outcome in terms of improved overall survival (OS), was presented at the Gastrointestinal Cancers Symposium in 2017. The same group subsequently reported that the levels of interlukein (IL)-8 and IL-15 were positively associated with the OS of pancreatic cancer patients, and served as potential predictive biomarkers for the therapeutic effect of combined OT-101 and chemotherapy administration for pancreatic cancer [25].

Signal transducer and activator of transcription (STAT) proteins are a family of cytoplasmic transcription factors that are thought to be candidates for anti-cancer therapeutic options, due to the higher dependency of cancer cells on STAT activity compared with normal cells [26]. Hong et al. showed that AZD9150, a STAT3-inhibiting ASO, has single-agent pharmacologic tumor-suppressive activity in models of human lymphoma and lung cancer [27]. They also reported the evaluation of AZD9150 in a non-Hodgkin’s lymphoma population [28]. A different group revealed the value of AZD9150 in combination with conventional chemotherapy for patients with neuroblastoma [29]. Programmed cell-death 1 ligand 1 (PD-L1) is a protein of the B7/CD28 family that controls T-cell activation, and MEDI4736, an antagonistic anti-PD-L1 monoclonal antibody, may be a promising therapeutic approach for the treatment of cancer [30]. Based on these studies, Hong and colleagues are now recruiting study patients to investigate whether AZD9150 administered intravenously in combination with MEDI4736 can control advanced pancreatic, lung, or colorectal cancer in a phase II clinical trial (NCT02983578).

X-linked inhibitor of apoptosis (XIAP), an endogenous caspase inhibitor, blocks the execution of apoptosis and is an important survival factor in cancer cells [31].

AEG-35156, a 19-mer phosphorothioate ASO targeting XIAP, was developed for the potential treatment of cancer by lowering the apoptotic threshold and inducing cell death, as well as enhancing the cytotoxic action of chemotherapeutic agents [32]. A preclinical study indicated that AEG35156 enhanced the sensitization of tumor necrosis factor (TNF)-related, apoptosis-inducing, ligand-mediated apoptosis in pancreatic carcinoma cells (Panc1) [33]. Then, a phase I trial of AEG35156 plus standard-dose gemcitabine administered intravenously in 14 patients with advanced pancreatic cancer was conducted, and showed good tolerability but no better results than with gemcitabine alone [34].

Heat shock protein 27 (Hsp27) is a chaperone implicated in several pathological processes; Hsp27 expression becomes highly upregulated, and is related to higher resistance to gemcitabine in patients with pancreatic cancer [35,36]. A modified ASO that is complementary to Hsp27 (OGX-427) has been developed, showing the inhibition of tumor progression and enhancement of gemcitabine’s efficacy in pancreatic cancer [37]. Thereafter, a phase II trial of OGX-427 plus gemcitabine and nab-paclitaxel administered intravenously in patients with metastatic pancreatic cancer was conducted, and revealed that the addition of OGX-427 to chemotherapy provided no improvement in the effect of the first-line setting [38]. Thus, clinical trials involving the ASO strategy as a new therapy, combined with conventional drugs for advanced states of pancreatic cancer, have not shown significant additional value for these patients, against our expectations.

## 4. SiRNA

SiRNA relies on sequence complementarity between RNAs and the target mRNAs to inhibit their activity. In particular, chemically synthesized siRNAs are mainly being used as a new class of therapeutic agents. The siRNA-based therapeutic strategy enables selective silencing of targeting gene expression. Compared with conventional anti-cancer drugs, siRNA has marked advantages, such as greater safety, stronger potency, higher specificity, and unrestricted choice of drug targets [39]. As a potent modality of silencing the specific expression of genes that contribute to tumorigenesis and poor prognosis in cancer [40], RNAi-mediated therapeutic intervention can be directed against pancreatic cancer through various pathways [41].

Protein kinase N3 (PKN3), a molecule distantly related to protein kinase C, is regulated by the phosphoinositide-3-kinase (PI3K) signal transduction pathway [42]. A preclinical study on Atu027, a liposomal siRNA molecule that specifically impedes PKN3 expression in a mouse model, highlights the further development of Atu027 as a novel siRNA formulation to block the progression of multiple solid cancers, including pancreatic cancer [43]. A first-in-human phase I trial of Atu027 confirmed its safety in four patients with advanced pancreatic cancer [44]. Subsequently, a clinical trial of Atu027 plus gemcitabine, administered intravenously in unresectable pancreatic cancer (NCT01808638), was planned in the NIH records, but the study has published no report as of yet.

Oncogenic *KRAS* mutation, an early event in the development of pancreatic cancer, is involved in over 90% of pancreatic cancer cases [45,46,47]. Pancreatic cancer cell growth is dependent on mutated *KRAS* activity [48]. Accordingly, silencing *KRAS* is indicative of the inhibition of pancreatic cancer proliferation [49]. On the basis of the above results, Silenseed Ltd. (Modi’in, Israel) developed LODER (Local Drug EluteR), and Zorde Khvalevsky et al. engineered the anti-KRAS^G12D^ siRNA (siG12D LODER). These demonstrate that LODER-derived siG12D significantly suppresses pancreatic cancer growth, both in vitro and in vivo, and impedes pancreatic tumor growth [50], although all of the past studies with direct *KRAS* modifications for pancreatic cancer had failed [51]. As siG12D LODER was designed to be directly implanted into pancreatic cancer under endoscopic ultrasonography (EUS) guidance, this strategy could resolve both the delivery problems associated with RNAi and the impermeable nature of the tumor microenvironment [52,53]. Golan et al. performed an open-label phase I/IIa clinical trial, and revealed that the combination of siG12D LODER and gemcitabine was well-tolerated and safe, and had potential value in 15 patients with locally advanced pancreatic cancer, as the median OS was 15.1 months and the 18-month survival rate was 38.5% [54]. Subsequently, the biocompatibility regarding the safety and toxicity of siG12D LODER as a local and prolonged delivery system for pancreatic cancer therapy was comprehensively assessed in rats after subcutaneous administration [55]. A phase II clinical trial of siG12D LODER in combination with gemcitabine plus nab-paclitaxel in patients with locally advanced pancreatic cancer has been ongoing (NCT01676259).

Exosomes, extracellular vesicles generated by all cells, are naturally present in circulating blood. Kamerkar et al. produced exosomes derived from normal fibroblast-like mesenchymal cells for the specific delivery of siRNA or shRNA to *KRAS^G12D^* (iExosomes), a common mutation in pancreatic cancer [56]. Their study showed that iExosomes repress tumor progression by augmenting the efficacy of exosomes in delivering the therapeutic payload in multiple mouse models of pancreatic cancer and significantly improved their OS, dependent on CD47-regulated exosome protection from phagocytosis by monocytes and macrophages [56]. The same group reported the bioreactor-based, large-scale production of clinical-grade exosomes, derived from bone marrow mesenchymal stem/stromal cells with a good manufacturing practices (GMP) grade, for promoting the clinical application of iExosome-based therapy [57]. Currently, a phase I study of mesenchymal stromal cell-derived iExosomes administered intravenously for metastatic pancreatic cancer patients harboring the *KRAS^G12D^* mutation has been planned (NCT03608631).

Recently, complex cancer cell-tumor stroma interactions have been focusing on their crucial roles in tumor initiation, progression, and metastasis [58]. Medicines targeting the tumor microenvironment, which mediates the therapeutic resistance of pancreatic cancer, have been progressing [59]. Glycosaminoglycan, a cancer stromal molecule, was reported to be involved in various cancer developmental phases, such as proliferation, invasion, metastasis, and angiogenesis [4,60,61,62]. Specifically, chondroitin sulfate-E (CS-E), a matrix glycosaminoglycan, was found to be expressed in both the tumor cells and stromal cells surrounding the tumor in pancreatic cancer patient tissues [61,63].

In agreement with the above evidence, we previously demonstrated that high CHST15 expression, which is responsible for the biosynthesis of sulfated CS-E, may represent a potential predictive marker of OS in patients with pancreatic cancer following surgical resection [64], and that STNM01, a CHST15 siRNA, successfully inhibited pancreatic tumor growth in xenograft experiments using the RNAi strategy, as shown in Figure 2 [4]. In an open-label trial, the safety and feasibility of STNM01 treatment under EUS-guided fine-needle injection (EUS-FNI) were certified in six patients with unresectable pancreatic cancer [65]. Based on this accumulating evidence, a clinical phase I/IIa study of STNM01 using EUS-FNI, for patients with unresectable pancreatic cancer, obtained approval from the Japan Agency for Medical Research and Development in 2018.

In short, an RNAi-based strategy for patients with pancreatic cancer is theoretically a prospective therapeutic option; therefore, further studies are anticipated to be progress to the clinic.

## 5. MicroRNA

The recent, wide-ranging progression of miRNA-based diagnostic and therapeutic approaches for cancer patients is encouraging, and has been widely reported, including in pancreatic cancer [66,67,68,69,70,71,72,73,74,75,76,77,78,79,80,81,82,83,84,85,86]. Intriguingly, the clinical applications have advanced mainly in diagnostic fields instead of with treatment options. Schultz et al. demonstrated the value of serum miRNAs as a diagnostic marker in patients with pancreatic cancer in a case-control study [87], and Su et al. subsequently identified the serum levels of co-expressed hub miRNAs as potential diagnostic and prognostic biomarkers for pancreatic cancer [88]. In particular, circulating microRNA-25 (miR-25) has a high specificity for pancreatic cancer, and can be used as a potential biomarker for its early detection, based on the comparative data of miR-25 with CEA and CA19-9, which are widely accepted and conventional tumor markers of pancreatic cancer [89,90]. Based on previous studies, a clinical trial of miR-25 for pancreatic cancer using a detection kit has been planned in China. If the efficacy of miR-25 as a diagnostic biomarker of pancreatic cancer is definitively proven in the trial, the clinical meaning for early detection would be significantly impactful. The biological properties of miRNAs that enable them to target multiple genes, and act as fine tuners of their in vivo expression, account for the simultaneous regulation of the related disease-specific genes in their network. Accordingly, clinical trials of miRNA for pancreatic cancer will probably continue to grow in number, especially in the field of diagnosis.

## 6. Aptamers

Aptamers are single-stranded oligonucleotide ligands that demonstrate a high affinity toward target proteins and inhibit their physiological effect, by forming complexes with proteins based on their unique three-dimensional (3D) folding in the later phase of disease progression [91,92]. The systematic evolution of ligands by exponential enrichment (SELEX) is an experimental procedure that enables researchers to choose the optimum aptamer from a vast RNA pool for binding with target proteins [93]. Aptamers have been reported to act as players of various roles, such as oncosuppressors, biomarkers, cargo, and detectors for pancreatic cancer, as shown in Table 3.

Kim et al. generated a 2’-fluoro-uridine modified RNA aptamer (P12FR2) that binds specifically to human pancreatic adenocarcinoma upregulated factor (PAUF), and showed the oncosuppressive role of P12FR2 both in vitro and in vivo, suggesting that P12FR2 is a potential therapeutic target of human pancreatic cancer [94]. Based on their previous work, PAUF is a novel secretory protein involved in pancreatic cancer progression [95].

Ray et al. devised an aptamer strategy to identify RNA ligands that was reported to detect structural differences between the secretomes of pancreatic cancer cells and noncancerous cells [96]. Within their identification strategy, an aptamer (M9-5) that preferentially bound the pancreatic cancer secretome over the noncancerous secretome and targeted cyclophilin B (CypB) was confirmed. The M9-5 binding analysis demonstrated excellent potential as a serum biomarker to distinguish between pancreatic cancer patients and healthy volunteers using the M9-5 aptamer [96]. The authors characterize the aptamer and the target protein, CypB. Their posttranslational modifications on CypB could discriminate CypB expressed in human cells versus bacteria, and their findings accelerated the use of aptamers as important biomarkers [97]. The same group reported a different targeted aptamer. Based on the evidence that most pancreatic cancers overexpress epidermal growth factor receptor (EGFR), a transmembrane receptor tyrosine kinase [98], the authors generated a nuclease-resistant RNA aptamer, and found that the aptamer was internalized by EGFR on human pancreatic cancer cells by efficiently delivering gemcitabine-containing polymers into EGFR-expressing cells, resulting in the in vitro inhibition of the proliferation of pancreatic cancer cells [99]. Their work supports the value of the aptamer strategy, in that the targeted drug delivery of active metabolites into cancer cells could effectively overcome chemoresistance and minimize toxic effects by decreasing uptake into normal cells.

Several studies have used SELEX aptamers for pancreatic cancer. Zhang et al. identified the usability of a SELEX aptamer (BC-15) as a detector of circulating tumor cells in the peripheral blood of patients with pancreatic cancer [100]. A different group successfully generated eight aptamers with high affinity for pancreatic ductal adenocarcinoma, using cell-SELEX targeting a human pancreatic ductal adenocarcinoma cell line, PL45 [101]. They further developed a truncated DNA aptamer, termed XQ-2d, and proved its high affinity and specificity against the PL45 pancreatic cancer cell line, with a detection ratio of 82.5% [102]. The group’s work might be a meaningful step in the development of probes as molecular imaging tools for the early detection of pancreatic cancer.

Direct and targeted delivery of effective treatment agents by means of aptamers to pancreatic cancer cells might maximize the therapeutic value and minimize possible adverse events by decreasing uptake into normal biological cells. A study showed that a pancreatic cancer-specific RNA aptamer can be used for the targeted drug delivery of the nucleoside drug 5-fluoro-2’-deoxyuridine to pancreatic cancer cells expressing alkaline phosphatase placental-like 2, a putative biomarker, leading to the inhibition of tumor growth [103].

Yoon et al. successfully developed a novel aptamer-based therapy for pancreatic cancer by upregulating the transcription factor CCAAT/enhancer-binding protein-α (C/EBPα), owing to its binding with pancreatic cancer-specific 2’-fluropyrimidine RNA aptamers P19 and P1, which were selected by SELEX, resulting in tumor-suppressive effects [104]. Subsequently, their group further improved the chemotherapeutic efficacy for pancreatic cancer by forming hybridized aptamer–drug conjugates that involved pancreatic cancer-specific RNA aptamer P19, conducive to higher selectivity of drug delivery with a less toxic effect [105]. Impressively, the authors proposed a different therapeutic conjugation of pancreatic cancer-specific 2’-fluoropyrimidine RNA-aptamers and C/EBPα-small activating RNA, which accounted for the inhibition of pancreatic cancer growth [106]. Their work suggests that diverse aptamer strategies can be applied for the treatment of pancreatic cancer, by focusing on the outstanding targeted drug delivery potency. In contrast, Kratschmer et al. provided a direct conjugation strategy, by generating small-molecule aptamer–toxin conjugates of auristatin, monomethyl auristatin E (MMAE), and monomethyl auristatin F (MMAF), and confirmed the improved cytotoxicity of MMAE and MMAF conjugated to an anti-transferrin receptor aptamer and an anti-EGFR aptamer in three different human pancreatic cancer cell lines [107].

Based on the evidence that the G protein-coupled cholecystokinin B receptor (CCKBR) is constitutively overexpressed on the surface of human pancreatic cancer cells, and plays a role in tumor cell proliferation [108], Clawson et al. identified and characterized high-affinity DNA aptamers to the CCKBR, and reported the efficacy of delivering diagnostic or therapeutic agents to pancreatic cancer cells with minimal side effects, using dual SELEX selection from a pool of multiple DNA aptamers [109]. Another group assessed the potential of pancreatic cancer stem cell-associated aptamers as a diagnostic marker and therapeutic option with a modified SELEX method [110]. The most recent study reported the therapeutic effects of an aptamer-based gemcitabine delivery system, APTA-12, on pancreatic cancer cells both in vitro and in vivo [111]. Altogether, the aptamer strategies have shown promise as diagnostic or therapeutic options for pancreatic cancer, mainly in the field of targeted drug delivery to cancerous lesions, based on the benefits of the SELEX procedure. Thus, it seems that a clinical trial of aptamer oligonucleotides for patients with pancreatic cancer should be planned in the near future.

## 7. Decoys

Decoys, which form a double-stranded DNA (dsDNA) structure, inhibit DNA transcription by blocking the activity of the dsDNA-binding transcription factors. Compared with the research on other oligonucleotides, the research on decoys for pancreatic cancer has lagged. Regarding the decoy methods used for pancreatic cancer, only one therapeutic and two delivery methods have been reported.

Cogoi et al. focus on myc-associated zinc-finger (MAZ), which is a G4-DNA structure that binds to nuclear proteins to promote the activation of *KRAS* transcription [112]. The *KRAS* mutation is a primary trigger in >90% of cases of pancreatic cancer progression [45,46,47], and contains a nuclear-hypersensitive element (NHE). Therefore, the authors designed oligonucleotides that mimic one of the G-quadruplexes formed by NHE (G4-decoys), and found that the decoy strategy inhibits *KRAS* in pancreatic cancer cells and reduces tumor growth in human pancreatic cancer xenograft mice [112]. The authors subsequently reported a lipid-modified strategy based on the use of palmitoyl-oleyl-phosphatidylcholine liposomes, which improved the delivery of the G4-decoy by coupling to the liposomes efficiently, resulting in decreased *KRAS* transcription levels and metabolic activity of pancreatic cancer cells [113].

Based on a previous study, which reported that high-mobility group AT-hook 1 (HMGA1) promotes chemoresistance to gemcitabine through an Akt-dependent mechanism in human pancreatic cancer cells [114], Hassan et al. introduce a new strategy of delivering decoy HMGA1-hyperbinding sites into the nucleus of pancreatic cancer cells using an engineered adenovirus, leading to the sequestration of excess amounts of nuclear HMGA1 with the decoy hyperbinding sites and resulting in the reduction of oncogenic pancreatic cells and HMGA1-associated gemcitabine resistance [115]. Taken together, decoy strategies are expected to be translated into various clinical applications, as a new cancer treatment to specifically target any overexpressed oncogenic transcription factor that binds double-stranded DNA.

Given the few preclinical studies with decoy oligonucleotides for pancreatic cancer, ample room remains for further development of clinical applications.

## 8. Clustered, Regularly Interspaced, Short Palindromic Repeats

More recently, clustered, regularly interspaced, short palindromic repeat (CRISPR)-directed gene editing technology has been successfully developed for knocking out the targeted gene [116]. Thus, we summarize several studies that have shown the value of CRISPRs in pancreatic cancer. Vorvis et al. identified that the FOXA2 transcription factor is involved in pancreatic cancer pathogenesis, by integrating gene and microRNA profiling analyses together with CRISPR/CRISPR-associated protein 9 (CRISPR/Cas9) genome editing technology [117]. The authors showed that FOXA2 knockout using CRISPR/Cas9 vectors in PANC-1 pancreatic cancer cells induced tumor growth in vivo [117]. Subsequently, Belvedere et al. revealed the role of miR-196a (a well-known oncogenic factor) as a function of Annexin A1 (a Ca^2+^-binding protein that is involved in pancreatic cancer development) in MIA PaCa-2 pancreatic cancer cells, obtained by an in vitro CRISPR/Cas9 genome editing system [118]. CRISPR has also been established as an important screening tool in the field of drug discovery. Steinhart et al. found that CRIPSR-based genetic screening technology, which enables the high-resolution detection of genetic vulnerabilities in cancer cells, identifies vulnerabilities in three independent, RNF43-mutant pancreatic cancer cell lines that rely on Wnt signaling for proliferation. The authors also identified the Wnt receptor Frizzled-5 as a common vulnerability that can be exploited therapeutically with antagonistic antibodies [119]. Based on the MEK signaling pathway, which is the major driver of tumor formation, and being a promising therapeutic approach in pancreatic cancer, Szlachta et al. validated genes whose depletion synergistically augments cellular sensitivity to MEK inhibition through large-scale in vivo and in vitro CRISPR knockout screens in pancreatic cancer cells. They also evaluated drug response using the in vivo CRISPR screening method [120]. In the most recent study, Bakke et al. used the genome-wide CRISPR screening approach to quantify gene-specific phenotypic variation in PANC-1 cells in response to gemcitabine, and confirmed that proteasome subunit alpha type-6 is an essential gene in pancreatic cancer cells [121]. Altogether, CRISPR-directed gene editing technology as a next screening or therapeutic approach to pancreatic cancer has considerable potential; therefore, further investigations regarding this method should be conducted in near future.

## 9. Conclusions

Recent advances in oligonucleotide therapeutics with specific and targeted inhibition of gene expression have been considerable, especially in the cancer field. Thus, this class of therapeutics is close to becoming a breakthrough cancer treatment. Compared with other cancers, the therapeutic effect of conventional small-molecule drugs or antibody medicines against pancreatic cancer is insufficient. Therefore, various molecular profiling strategies for pancreatic cancer have been elucidated by the latest generations of genomics, transcriptomics, and proteomics approaches. Oligonucleotide therapeutics can match this genomic profiling trend and target the molecules, including non-coding RNAs, which were not therapeutic targets previously, enabling the targeting of the *KRAS* gene, a critical trigger of pancreatic cancer progression. The possibility of applying oligonucleotide therapeutics in the regulatory mechanisms of the microenvironment of pancreatic cancer has been indicated. Continuous inquiry into efficient delivery systems for oligonucleotide therapeutics remains an obstacle. Although the EUS-FNI procedure can be effective in treatment with oligonucleotides for local pancreatic cancer, the necessity of a next-generation delivery system is still needed for the metastatic state. Depending on the results of current oligonucleotide therapeutics clinical trials, the safety, efficacy, and proper selection of treatment can be better understood. Due to the progression of research on the modality, a new definitive treatment against refractory disease could be developed in the future.

## Figures and Tables

**Figure 1 ijms-20-03331-f001:**
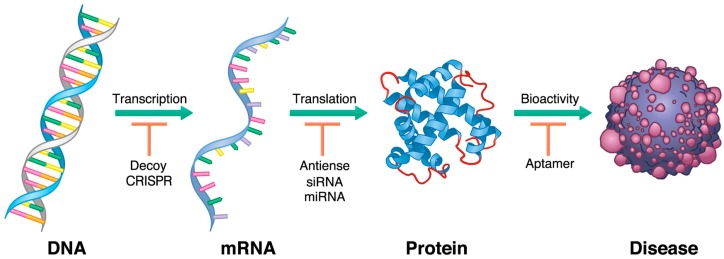
Therapeutic oligonucleotides act on different stages of pathological gene expression. Standard schematic of oligonucleotide activities in pathogenesis disease progression. Decoys bind to transcription factors of targeted DNA at the earliest phase. Subsequently, antisense RNA, small interfering RNA (siRNA), and microRNA (miRNA) act to target mRNAs. Then, aptamers directly inhibit proteins in the process of pathogenesis.

**Figure 2 ijms-20-03331-f002:**
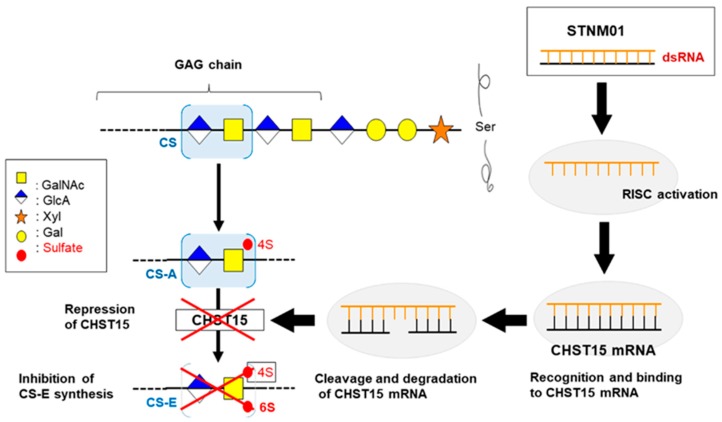
STNM01 blocks the glycosaminoglycan synthesis pathway through RNA-induced silencing complex (RISC) activation. Chondroitin sulfate (CS)-E is a matrix glycosaminoglycan (GAG), a linear polysaccharide composed of a repeating disaccharide unit containing D-glucuronic acid (GlcA) and N-acetyl-D-galactosamine (GalNAc), whose 4- and 6-positions are sulfated. Carbohydrate sulfotransferase 15 (CHST15) is a type-2 transmembrane Golgi protein that transfers sulfate to position 6 of GalNAc (4SO4) residues of CS-A to yield CS-E.

**Table 1 ijms-20-03331-t001:** Clinical trials of Oligonucleotide therapeutics for patients with pancreatic cancer.

Trial Identifer	The Name of Tested Oligonucleotide	Target Molecule	Category of Agent	Enrollment	Organizing Location	Study Phase
NCT00005594	ISIS 2503	Hras	Antisense	48	United States	Phase 2
NCT00844064	AP 12009/OT-101	TGF-β2	Antisense	37	Germany	Phase 1/2
NCT02983578	AZD9150	STAT3	Antisense	75 (estimated)	United States	Phase 2
NCT00557596	AEG35156	XIAP	Antisense	14	United States	Phase 1
NCT01844817	OGX-427/apatorsen	Hsp27	Antisense	132	United States	Phase 2
NCT01808638	Atu027	PKN3	siRNA	29	Germany	Phase 1/2
NCT01188785	siG12D LODER	KrasG12D	siRNA	15	United States	Phase 1
NCT01676259	siG12D LODER	KrasG12D	siRNA	80 (estimated)	United States	Phase 2
NCT03608631	iExosomes	KrasG12D	siRNA	28 (estimated)	United States	Phase 1
NCT03432624	Detection Kit	MiR-25	miRNA	750 (estimated)	China	

TGF-β2: transforming growth factor-beta 2; XIAP: X-Linked Inhibitor of Apoptosis; PKN3: protein kinase N3; Hsp27: heat shock protein 27.

**Table 2 ijms-20-03331-t002:** Classification of oligonucleotide therapeutics.

	Antisense	siRNA	miRNA	Aptamer	Decoy	CRISPR
**Structure**	ssDNA/RNA	dsRNA	dsRNA, shRNA	ssDNA/RNA	dsDNA	sgRNA
**Target**	mRNA, miRNA	mRNA	mRNA	Protein	Protein	dsDNA
	Pre-mRNA				(transcription factor)	
**Active site**	Intracellular	Intracellular	Intracellular	Extracellular	Intracellular	Intracellular
**Action**	mRNA decay		miRNA	Functional	Transcriptional	Adaptive
**Mechanism**	Splicing inhibition	mRNA decay	Complement	Inhibition	Inhibition	Immunity
	miRNA inhibition					

ssDNA/RNA: Single-stranded DNA/RNA, dsRNA: double-stranded RNA, shRNA: small hairpin RNA, sgRNA: single-guide RNA.

**Table 3 ijms-20-03331-t003:** Current studies of aptamer for pancreatic cancer.

No.	Target Agent	How to Work?	Ref.
1	P12FR2	Oncosuppressor	[94]
2	C/EBPα-saRNA	Oncosuppressor	[106]
3	cyclophilin B	Biomarker	[96,97]
4	circulating tumor cells	Biomarker	[100]
5	EGFR	Targeted delivery	[99]
6	ALPPL2	Targeted delivery	[103]
7	ApDCs P19	Targeted delivery	[105]
8	Auristatin-Modified Toxins	Targeted delivery	[107]
9	CCKBR	Targeted delivery	[109]
10	APTA-12	Targeted delivery	[111]
11	XQ-2d	Detector	[102]
12	cancer stem cells	Detector	[110]

P12FR2: 2-fluoropyrimidine modified RNA aptamer; SELEX: systematic evolution of ligands by exponential enrichment; C/EBPα: CCAAT/enhancer-binding protein-α; EGFR: epidermal growth factor receptor; ALPPL2: alkaline phosphatase placental-like 2; ApDCs: Aptamer-drug conjugates; CCKBR: cholecystokinin B receptor; APTA-12: a gemcitabine-incorporated AS1411; XQ-2d: a truncated DNA aptamer.

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
