# Peer review of "The Clinical Potential of Oligonucleotide Therapeutics against Pancreatic Cancer"

_ijms, 2019, doi:10.3390/ijms20133331_

Round 1
Reviewer 1 Report
The authors approached to summarize the current state of research on the application of therapeutic oligonucleotides in treatment of pancreatic cancer. The topic is important, especially in the light of limited treatment options of this type of malignant tumors. However, the present version of the manuscript has many drawbacks and I do not recommend its publication.
The language is confusing, many terms are used incorrectly or imprecisely. The text, after rewriting should be checked by native English speaker familiar with the subject of the review. Below are few examples selected from among many parts of the text requiring correction:
Page 1, line 12: “Although there is a several array of diagnostic and therapeutic choices for pancreatic cancer in recent years…”, please consider ““Although an array of diagnostic and therapeutic modalities for pancreatic cancer was proposed in recent years there is still an urgent need for improved therapeutic strategies”.
Page 1, line 14: “…antisense RNAs, RNA interference, aptamers and decoys…” Though antisense type of interactions between therapeutic oligonucleotide (TO) and its nucleic acid target is common for different types of TOs, antisesne oligonucleotides (ASO) are essentially short pieces of DNA (not RNA). The ASO is termed an "antisense" because its base sequence is complementary to the messenger RNA (mRNA), which is called the "sense" (coding) strand. The formed heteroduplex DNA/RNA is a substrate for RNase H which digests RNA part (but not DNA) within the heteroduplex preventing translation of the gene. Another type of AOS mechanism of action is found for steric-blocker oligonucleotides acting via “translation arrest” (without involvement of RNase H), in such case AOS can be RNA or DNA oligonucleotide (usually heavy modified).
“RNA interference” is not a type of TO, but a biological process, the active agents in RNA interference (and potential TOs) are miRNA and siRNA.
Page 1, line 29: “Oligonucleotide therapeutics is a general term for state-of-the-art molecular-targeted agents that have chain-like structures of several nucleic acids…”. What do the authors understand by “….have chain-like structures of several nucleic acids…”? Please provide commonly used definition of TOs.
Page 7, Table 2 and in the text: the term “target molecule” should be used consistently. In the drug design area the “target” molecule is usually an entity, often protein to which the drug molecule should bind preferentially inducing therapeutic effect. In diagnostics, the target molecule is the molecule to which the reporter molecule binds helping to detect it. Authors use the term “target molecule” in both meanings which causes confusion. In addition, “circulating tumor cells” or “cancer stem cells” are not molecules (Table 2).
Author Response
Point 1: Page 1, line 12: “Although there is a several array of diagnostic and therapeutic choices for pancreatic cancer in recent years…”, please consider ““Although an array of diagnostic and therapeutic modalities for pancreatic cancer was proposed in recent years there is still an urgent need for improved therapeutic strategies”.
Response 1: Thank you for the comment. I apologize that my English is not enough, though this paper was checked by the native speaker at American Journal Experts before my submission. Anyway, I agreed to change the phrase to be better.
Point 2: Page 1, line 14: “…antisense RNAs, RNA interference, aptamers and decoys…” Though antisense type of interactions between therapeutic oligonucleotide (TO) and its nucleic acid target is common for different types of TOs, antisense oligonucleotides (ASO) are essentially short pieces of DNA (not RNA). The ASO is termed an "antisense" because its base sequence is complementary to the messenger RNA (mRNA), which is called the "sense" (coding) strand. The formed heteroduplex DNA/RNA is a substrate for RNase H which digests RNA part (but not DNA) within the heteroduplex preventing translation of the gene. Another type of ASO mechanism of action is found for steric-blocker oligonucleotides acting via “translation arrest” (without involvement of RNase H), in such case ASO can be RNA or DNA oligonucleotide (usually heavy modified).
“RNA interference” is not a type of TO, but a biological process, the active agents in RNA interference (and potential TOs) are miRNA and siRNA.
Response 2: Thank you for the comments. I apologize for my confusing description about RNAi. I modified from using “RNAi” to “siRNA” as a therapeutic options.
Point 3: Page 1, line 29: “Oligonucleotide therapeutics is a general term for state-of-the-art molecular-targeted agents that have chain-like structures of several nucleic acids…”. What do the authors understand by “….have chain-like structures of several nucleic acids…”? Please provide commonly used definition of TOs.
Response 3: Thank you for the comments. I changed my explanation about TOs.
Point 4: Page 7, Table 2 and in the text: the term “target molecule” should be used consistently. In the drug design area the “target” molecule is usually an entity, often protein to which the drug molecule should bind preferentially inducing therapeutic effect. In diagnostics, the target molecule is the molecule to which the reporter molecule binds helping to detect it. Authors use the term “target molecule” in both meanings which causes confusion. In addition, “circulating tumor cells” or “cancer stem cells” are not molecules (Table 2).
Response 4: Thank you for the comments. I apologize for my confusing description about target molecule. Considering the above, I modified appropriately in Table 2 and in the Aptamers chapter.

Reviewer 2 Report
The authors present a review in the use of oligonucleotide therapeutics as promising agents against pancreatic cancer and summarize the current status of clinical trials using this approach. The authors attempt to describe some of the underlying preclinical or scientific data with correlations to clinical implications.
The authors are clearly competent describing all oligonucleotides might be effective against pancreatic cancer and are equally competent in describing the current progress in this field. But there are very exciting things happening with single-stranded oligonucleotides as donor DNA and CRISPR-directed gene editing. This field will overwhelmingly subsume most of the corollary approaches. Thus, the authors need to discuss how oligonucleotides could be used with programmable nucleases in much greater detail since it is likely the combinatorial therapy will win out.
The authors should also consider how best clinical translation takes place, utilizing oligonucleotides alone or in the combinatorial approach described above. Will their proposed therapeutic agent have a sustained effect on the patients that will truly make a difference in the refractory outcomes? What experience has been learned from the early transition to a clinical application, good and bad? A greater emphasis on this is needed.
The authors need also to expand upon their discussion regarding delivery which is been the most difficult and vexing problem for all DNA medicines.
These are certainly topics that can be incorporated into the next version. I suggest resubmission of the revised manuscript addressing all points above prior to acceptance for publication.
Author Response
Point 1: The authors are clearly competent describing all oligonucleotides might be effective against pancreatic cancer and are equally competent in describing the current progress in this field. But there are very exciting things happening with single-stranded oligonucleotides as donor DNA and CRISPR-directed gene editing. This field will overwhelmingly subsume most of the corollary approaches. Thus, the authors need to discuss how oligonucleotides could be used with programmable nucleases in much greater detail since it is likely the combinatorial therapy will win out.
Response 1: I am thankful for the guidance about CRISPR-directed gene editing technology. I agree with that the field might be a crucial approach against pancreatic cancer and add the chapter in my paper.
Point 2: The authors should also consider how best clinical translation takes place, utilizing oligonucleotides alone or in the combinatorial approach described above. Will their proposed therapeutic agent have a sustained effect on the patients that will truly make a difference in the refractory outcomes? What experience has been learned from the early transition to a clinical application, good and bad? A greater emphasis on this is needed.
The authors need also to expand upon their discussion regarding delivery which is been the most difficult and vexing problem for all DNA medicines.
Response 2: I agree with the comments. I added the explanation about the route of administration for each trials and modified my conclusions. But I don’t touch about the delivery problem with the intention of just focusing on the clinical potential of oligonucleotide therapeutics in pancreatic cancer, based on the existence of several review articles regarding drug delivery matter.

Reviewer 3 Report
The manuscript by Takakura et. al. summarizes oligonucleotide therapeutics against pancreatic cancer that have reached clinical trials. The topic is certainly timely and interesting but the presentation unfortunately leaves a lot to be desired. The manuscript could become valuable reference material for people working in the hot field of oligonucleotide therapeutics but as it stands it just feels unfinished. The following problems have to be taken care of before publication can be considered:
There are many issues with the language, ranging from minor incongruences, such as “…is a several array…” on page 1, line 12 to impenetrable sentences such as “Since then…” on page 2, line 23. Thorough proofreading by a native English speaker is strongly recommended.
The conclusions paragraphs serves little purpose as it is, consisting of three redundant sentences. The state of the art, major breakthroughs expected in near future and main challenges to overcome should be summarized much more explicitly.
What is meant by “gene expression in oligonucleotide” on page 1, line 35?
The description of RNAi as “a breakthrough application” on page 2, line 22 is odd as RNAi is a naturally occurring phenomenon and not an application.
The statement on page 4, line 28 that “RNAi relies on sequence complementarity between RNAs and the target mRNA” is, of course true, but not characteristic of just RNAi. Antisense oligonucleotides also bind to a complementary sequence within mRNA.
The termn “LODER” on page 4, line 49, should be defined.
The sentences on page 7, line 24 (“Several studies…”) and page 7, line 26 (“A different group…”) seem to counterpose “SELEX aptamers” with “DNA aptamer” even though also the latter (and indeed all aptamers) are, in fact, obtained through a SELEX process.
The importance of KRAS mutation in progression of pancreatic cancer is mentioned twice, on page 4, line 45 and page 8, line 50, with different literature references.
Author Response
Point 1: There are many issues with the language, ranging from minor incongruences, such as “…is a several array…” on page 1, line 12 to impenetrable sentences such as “Since then…” on page 2, line 23. Thorough proofreading by a native English speaker is strongly recommended.
Response 1: Thank you for the comment. I apologize that my English is not enough, though this paper was checked by the native speaker at American Journal Experts before my submission. Anyway, I will ask thorough proofreading by IJMS English speaker.
Point 2: The conclusions paragraphs serves little purpose as it is, consisting of three redundant sentences. The state of the art, major breakthroughs expected in near future and main challenges to overcome should be summarized much more explicitly.
Response 2: Thank you for the comment. I modified the conclusions paragraph.
Point 3: What is meant by “gene expression in oligonucleotide” on page 1, line 35?
Response 3: Thank you for the comments. I apologize for my confusing description “gene expression in oligonucleotide” on page 1, line 35 and delete the part.
Point 4: The description of RNAi as “a breakthrough application” on page 2, line 22 is odd as RNAi is a naturally occurring phenomenon and not an application.
Response 4: Thank you for the comments. I apologize for my confusing description about RNAi. I modified from “application of oligonucleotides” to “biological process”.
Point 5: The statement on page 4, line 28 that “RNAi relies on sequence complementarity between RNAs and the target mRNA” is, of course true, but not characteristic of just RNAi. Antisense oligonucleotides also bind to a complementary sequence within mRNA.
Response 5: Thank you for the comments. I modified here.
Point 6: The term “LODER” on page 4, line 49, should be defined.
Response 6: Thank you for the comments. I added the explanation.
Point 7: The sentences on page 7, line 24 (“Several studies…”) and page 7, line 26 (“A different group…”) seem to counterpose “SELEX aptamers” with “DNA aptamer” even though also the latter (and indeed all aptamers) are, in fact, obtained through a SELEX process.
Response 7: Thank you for the comments. I agree that these parts should not be counterpose. So I delete a part.
Point 8: The importance of KRAS mutation in progression of pancreatic cancer is mentioned twice, on page 4, line 45 and page 8, line 50, with different literature references.
Response 8: Thank you for the comments. I modified Refs. Appropriately.

Reviewer 4 Report
In the review Takakura et al. collected data on clinical trials of therapeutic oligonucleotides used for treatment of patients with pancreatic cancer. Although in general the topic of the review is interesting and will be useful for readers working in this area of medicinal chemistry, the content and composition require major improvements to be acceptable for publication. My major concerns are as follows:
1. The description of the use of nucleic acid-backbone oligomers as therapeutics is highly imprecise. The authors describe oligonucleotide therapeutics as antisense ASO oligonucleotides and mix up antisense oligodeoxyribonucleotides with antisense RNA. First of all, it should be mentioned that oligonucleotides which recognize complementary sequences of messenger RNA (mRNA) by the Watson-Crick hydrogen bonding are generally named “antisense”. Their activity is based either on mRNA cleavage or hybridization arrest, and in this way they alter gene expression. Depending on the applied therapeutic oligonucleotide, the RNA cleavage may occur within various mechanisms. (i) antisense oligodeoxyribonucleotide ASO (plural: ASOs) binds to mRNA and the formation of a DNA/RNA duplex activates RNase H which cleaves the RNA strand leading to mRNA degradation; (ii) the binding of a guide strand of siRNA to the complementary fragment of target mRNA activates the Ago nuclease present in the RISC complex and results in mRNA degradation; (iii) some other antisense mRNA-based inhibitory molecules can be listed here, which exert RNA backbone cleavage via catalytic activity as ribozymes and deoxyribozymes. Also ASOs targeted towards pathogenic miRNA are known (called antimirs). Besides, triplex forming oligonucleotides (oligodeoxyribonucleotides binding to the double stranded DNA according to Hoogsteen or reverse Hoogsteen mode) which are able to block the DNA function on the transcription step. The review requires corrections thorough the text.
2. It is not true that therapeutic oligonucleotides are obtained only by “chemosynthetic” methods. Some of nucleic acid therapeutics serve as synthetic oligonucleotides (ASO, siRNA), but some as shRNA can be expressed from the DNA plasmids. Correct the description on page 1, lane 32-35.
3. Another approach is represented by therapeutic oligonucleotides which exert inhibitory activity towards selected proteins, and here aptamers obtained in the SELEX process can be mentioned, as well as synthetically obtained decoy oligodeoxyribonucleotides. The latter in the form of corresponding duplex (decoy ODNs), bind to the target transcription factors and inactivate them in the transcription process. Therefore, the message that therapeutic nucleic acids are active downstream of the gene expression level is not true. Please, rephrase the text.
4. The authors might mention the CRISPR technique for inhibition of expression of genes associated with cancer, and present preclinical trials for slowdown the pancreatic cancer.
5. I am not sure if clinical trials to prove the usefulness of a given miRNA as a pancreatic cancer marker can be included to the therapeutic oligonucleotides approach. However, such a chapter can be included into the review being evaluated following a clear explanation of the approach the miRNA is recognized (e.g. by the ASO). Anyway, extensive citation of approaches for the use of miRNA as therapeutic target (ref.64-84) should be accompanied by more detailed description of their perspectives to become therapies.
6. Another issue which seems to be misunderstood by the authors is the “cancer progression” identified with the step the gene expression is inhibited (at the levels of transcription, translation or protein activity, as it is shown at Figure 1, see also p.6 line 25; p.8 line 44). Please, make the respective corrections along the text and figures.
7. Re the paragraph on ASO: this part does not contain statistical data describing therapeutic efficacy of antisense oligodeoxyribonucleotides (ASOs).
8. Section 8 – Conclusions is a brief summary only; it does not indicate the best candidate among therapeutic oligonucleotides and a best drug carrier in therapy of pancreatic cancer based on a rich cited literature.
9. The NCT number of each clinical trial would be helpful to verify the data.
10. The other comments and corrections needed to be introduced into the text:
- In Table 1 the authors mention “targeted drug” which is complete misunderstanding the meaning of these data. It should be “Abbreviated name of the tested oligonucleotide”. Please, insert the references.
- Explain sentence p.4 lines 34-35.
- Reword: resistibility, orthotopic, xenograft experiments,
- Scheme 2 is not clear, what does the term “risc activation” mean? How the cleavage of mRNA is generated?
- Table 2, the nature of the aptamer is not assigned (DNA or RNA backbone). The Table caption should be rephrased.
- Page 9, line 1-3, not clear combination of the G4 aptamer and decoy strategies, needs some explanation. The description of the decoy approach given on p.9 lines 7-13 is also too general and unclear.
- Does ref 5 really describe the first antisense????? I recommend to remove this reference.

Author Response
Point 1: The description of the use of nucleic acid-backbone oligomers as therapeutics is highly imprecise. The authors describe oligonucleotide therapeutics as antisense ASO oligonucleotides and mix up antisense oligodeoxyribonucleotides with antisense RNA. First of all, it should be mentioned that oligonucleotides which recognize complementary sequences of messenger RNA (mRNA) by the Watson-Crick hydrogen bonding are generally named “antisense”. Their activity is based either on mRNA cleavage or hybridization arrest, and in this way they alter gene expression. Depending on the applied therapeutic oligonucleotide, the RNA cleavage may occur within various mechanisms. (i) antisense oligodeoxyribonucleotide ASO (plural: ASOs) binds to mRNA and the formation of a DNA/RNA duplex activates RNase H which cleaves the RNA strand leading to mRNA degradation; (ii) the binding of a guide strand of siRNA to the complementary fragment of target mRNA activates the Ago nuclease present in the RISC complex and results in mRNA degradation; (iii) some other antisense mRNA-based inhibitory molecules can be listed here, which exert RNA backbone cleavage via catalytic activity as ribozymes and deoxyribozymes. Also ASOs targeted towards pathogenic miRNA are known (called antimirs). Besides, triplex forming oligonucleotides (oligodeoxyribonucleotides binding to the double stranded DNA according to Hoogsteen or reverse Hoogsteen mode) which are able to block the DNA function on the transcription step. The review requires corrections thorough the text.
Response 1: Thank you for the instructive comments. In my paper, our aim is to demonstrate the clinical potentials of oligonucleotides against pancreatic cancer, therefore I just want to explain the types of oligonucleotides. For contrasting with the other oligonucleotides, I need to lump together as ASOs. But, I agreed with the mechanisms of oligonucleotides are different variously depend on the applied agents. So I just want to write together as a therapeutic candidates from a clinical standpoint.
Point 2: It is not true that therapeutic oligonucleotides are obtained only by “chemosynthetic” methods. Some of nucleic acid therapeutics serve as synthetic oligonucleotides (ASO, siRNA), but some as shRNA can be expressed from the DNA plasmids. Correct the description on page 1, lane 32-35.
Response 2: Thank you for the comments. I corrected the part.
Point 3: Another approach is represented by therapeutic oligonucleotides which exert inhibitory activity towards selected proteins, and here aptamers obtained in the SELEX process can be mentioned, as well as synthetically obtained decoy oligodeoxyribonucleotides. The latter in the form of corresponding duplex (decoy ODNs), bind to the target transcription factors and inactivate them in the transcription process. Therefore, the message that therapeutic nucleic acids are active downstream of the gene expression level is not true. Please, rephrase the text.
Response 3: Thank you for the comments. At least in current clinical trials for pancreatic cancer, these approaches are similar to inhibit the activity of specific targeted gene. But, I agreed with the existence of several approaches of therapeutic oligonucleotides and rephrased my paper properly.
Point 4: The authors might mention the CRISPR technique for inhibition of expression of genes associated with cancer, and present preclinical trials for slowdown the pancreatic cancer.
I am not sure if clinical trials to prove the usefulness of a given miRNA as a pancreatic cancer marker can be included to the therapeutic oligonucleotides approach. However, such a chapter can be included into the review being evaluated following a clear explanation of the approach the miRNA is recognized (e.g. by the ASO). Anyway, extensive citation of approaches for the use of miRNA as therapeutic target (ref.64-84) should be accompanied by more detailed description of their perspectives to become therapies.
Response 4: Thank you for the comments. The reason why the title is not therapeutic potential of oligonucleotides,,,,but clinical potential,,,is due to the diagnostic value of oligonucleotides in pancreatic cancer treatment.
Point 5: Another issue which seems to be misunderstood by the authors is the “cancer progression” identified with the step the gene expression is inhibited (at the levels of transcription, translation or protein activity, as it is shown at Figure 1, see also p.6 line 25; p.8 line 44). Please, make the respective corrections along the text and figures.
Response 5: Thank you for the comments. I agree. Figure 1. is just presented for helping the understanding of standard acting of oligonucleotides in several phases of disease progression, not in cancer progression.
Point 6: Re the paragraph on ASO: this part does not contain statistical data describing therapeutic efficacy of antisense oligodeoxyribonucleotides (ASOs).
Response 6: Thank you for the comments. Unfortunately, as far as I searched, there is no study regarding ASO against pancreatic cancer with statistical analysis.
Point 7: Section 8 – Conclusions is a brief summary only; it does not indicate the best candidate among therapeutic oligonucleotides and a best drug carrier in therapy of pancreatic cancer based on a rich cited literature.
Response 7: Thank you for the comments. I modified the conclusions.
Point 8: The NCT number of each clinical trial would be helpful to verify the data.
Response 8: Thank you for the comments. All of the related NCT numbers are in Table 1.
Point 9: The other comments and corrections needed to be introduced into the text:
- In Table 1 the authors mention “targeted drug” which is complete misunderstanding the meaning of these data. It should be “Abbreviated name of the tested oligonucleotide”. Please, insert the references.
- Explain sentence p.4 lines 34-35.
- Reword: resistibility, orthotopic, xenograft experiments,
- Scheme 2 is not clear, what does the term “risc activation” mean? How the cleavage of mRNA is generated?
- Table 2, the nature of the aptamer is not assigned (DNA or RNA backbone). The Table caption should be rephrased.
- Page 9, line 1-3, not clear combination of the G4 aptamer and decoy strategies, needs some explanation. The description of the decoy approach given on p.9 lines 7-13 is also too general and unclear.
- Does ref 5 really describe the first antisense? I recommend to remove this reference.
Response 9: Thank you for the comments.

Round 2
Reviewer 3 Report
The manuscript has been substantially improved and can be accepted for publication. Some language issues remain but those can be taken care of during copy-editing.
Author Response
The manuscript has been substantially improved and can be accepted for publication. Some language issues remain but those can be taken care of during copy-editing.
Thank you for the comments.
I asked English modification to MDPI English editing in addition to native check by American Journal Experts. I attached the certificate.

Reviewer 4 Report
The comments are given in the attached file.

Author Response
The review by Takakura et al. deals with collection of data on the clinical trials of therapeutic oligonucleotides for treatment of patients with pancreatic cancer. Although in general the topic of the review is interesting and will be useful for readers working in this area of medicinal chemistry, the text of the review and its composition require major improvements to be acceptable for publication. My major concerns deal with the following topics:
1. The description of the use of nucleic acid-backbone oligomers as therapeutics is far from the requested. The authors with limited precision describe oligonucleotide therapeutics as antisense ASO oligonucleotides. They mix up the antisense oligodeoxyribonucleotides with antisense RNA. Please, define antisense oligonucleotides ASOs as oligodeoxyribonucleotides.
In such a review a clear definition of each approach should be properly explained. First of all it is worth to mention that oligonucleotides which recognize the complementary sequence of the messenger RNA (mRNA) by the Watson-Crick hydrogen bonding (means by binding to complementary RNA sequence) are generally termed as “antisense”. Their activity is based on the exertion either mRNA cleavage or hybridization arrest, and in this way they contribute to altering the gene expression. Depending on the applied therapeutic oligonucleotide the RNA cleavage may occur within various mechanisms. (i) antisense oligodeoxyribonucleotide ASO (plural: ASOs) binds to mRNA and the formation of a DNA/RNA duplex activates RNase H which cleaves the RNA strand leading to mRNA degradation; (ii) binding of the guide strand of siRNA to complementary fragment of target mRNA activates the Ago nuclease present in RISC complex and results in mRNA degradation; (iii) some other antisense mRNA-based inhibitory molecules can be listed here which exert RNA backbone cleavage via catalytic activity as ribozymes and deoxyribozymes. Also ASOs targeted towards pathogenic miRNA are known (called antimirs). Besides, triplex forming oligonucleotides (oligodeoxyribonucleotides binding to the double stranded DNA according to Hoogsteen or reverse Hoogsteen mode) which are able to block the DNA function on the transcription step. The review requires corrections thorough the text. This is not done at all. Moreover, antisense oligonucleotides do not recognize “nucleotide sequence” but mRNA fragment of a given sequence. Please, correct p.1/18. Besides, remove p.4/32 “like the other oligonucleotides”. Each oligonucleotides exerts its gene silencing activity by different mechanism. Again, there is nothing mentioned in the review!
2. It is not true that therapeutic oligonucleotides are obtained only by “chemosynthetic” methods. Some of nucleic acid therapeutics serve as synthetic oligonucleotides (ASO, siRNA), but some as shRNA can be expressed from the DNA plasmids. Correct the description on page 1, lane 32-35.
3. The other approach is represented by therapeutic oligonucleotides which can exert inhibitory activity towards selected proteins, and here aptamers obtained in the SELEX process can be mentioned. Another type are decoy oligodeoxyribonucleotides, which are synthetically obtained oligomers of a complementary sequence, being fragments of the genomic DNA recognized by the transcription factors. Such decoy oligodeoxyribonucleotide duplexes (decoy ODNs) can bind to the target transcription factors and inactivate the factors in the transcription process. Therefore, the expression used in the text that therapeutic nucleic acids are active downstream of the gene expression level is not true. Please, rephrase the text. The authors should be more precise talking about DNA and RNA -decoy oligomers (“the first clinical trial related to a decoy strategy was performed for HIV-1 infection, as with aptamers [13]” p.2/29 versus p.8/42 when they describe DNA decoys).
4. The authors might mention the CRISPR technique for inhibition of expression of genes associated with cancer, and focus of preclinical trials for slowdown the pancreatic cancer.
5. I am not sure if clinical trials aiming to prove the usefulness of a given miRNA as a pancreatic cancer marker can be included to the therapeutic oligonucleotides approach. However, such a chapter can be included into the review under consideration of a clear explanation of the approach the miRNA is recognized (can be e.g. by the ASO). Anyway, extensive citation of approaches for the use of miRNA as therapeutic target (ref.64-84) should be accompanied by more deep description for the perspectives to be used as drugs.
6. Another issue which seems to be misunderstood by the authors is the “cancer progression” identified with the step the gene expression is inhibited (on the level of transcription, translation or on the level of protein activity, as it is shown at Figure 1, see also p.6 line 25; p.8 line 44). Please, make the respective corrections along the text and figures. This point is still not fully corrected since there are at least several cases when “disease progression” is missed with upstream or downstream expression. Please, correct.
7. Re the paragraph on ASO: this part does not contain statistical data describing therapeutic efficacy of antisense oligodeoxyribonucleotides (ASOs).
8. Section 8 – Conclusions it is a brief summary only; it does not describe the best candidate among therapeutic oligonucleotides and a best drug carrier in therapy of pancreatic cancer based on a rich cited literature.
9. The NCT number of each clinical trial would be helpful to verify the data.
10. The other comments and corrections needed to introduce into the text:
- In Table 1 the authors mention “targeted drug” which is complete misunderstanding the meaning of these data. It should be “Abbreviated name of the tested oligonucleotide”. Please, insert the references. Not corrected!!!!!
- Explain sentence p.4 lines 34-35.
- Reword: resistibility, orthotopic, xenograft experiments,
- Scheme 2 is not clear, what does “risc activation” mean? How cleavage of mRNA is generated? Not done
- Table 2, the nature of aptamer is not assigned (DNA or RNA backbone). The Table caption should be rephrased.
- Page 9, line 1-3, not clear combination of aptamer and decoy strategy. Decoy approach description given on p.9 lines 7-13 is also too general and unclear.
- Is ref 5 really the first antisense????? I recommend to remove this reference. Please remove this reference since it describes the synthesis of dinucleoside phosphate fragment of DNA and not an antisense strategy for inhibition of gene expression. Eventually you may explain why you cite it, what idea is suggested there (citing the text).
I sincerely appreciate your fruitful suggestions.
Because we basically agreed with your comments, I revised the relevant parts according to your kind instruction. In addition, I asked English modification to MDPI English editing. I attached the certificate.
As you pointed out, we also recognize that the difference of action mechanisms is important in the field. But, I am so sorry for your confusion regarding my descriptions. Our concept for this review article is mainly focusing on how to use oligonucleotides for pancreatic cancer both in clinical and pre-clinical so far. Because, there has not been such a review paper related to the topic until now.
Therefore, we ask for your kind understanding that Chapter 2 (Effects of oligonucleotide therapeutics on disease progression) is positioned as not a chapter focusing on the action mechanism itself of oligonucleotides, but a chaper briefly mentioning about history of oligonucleotide therapeutics (from pioneer works to current diversity), based on the existence of several review papers that are focusing on the action mechanism of oligonucleotides. To clarify our concept, I changed the title of chapter 2.
But meanwhile, we added a new table regarding the difference of action mechanisms of each oligonucleotides, in addition, we explained as please kindly refer previous papers for the greater level of detail with Refs, because I understand its importance.
General comment:
1. The statement “Targeted drug delivery” is OK for expression “targeted delivery of drug to the target cells”. However, you use wrong expression “targeted drug” – is you want to indicate that the drug is targeting the gene/protein the term “targeting drug” is a proper name. Please, correct along the text.
Thank you for the comments. I revised all of the relevant parts according to your instruction.
2. The fragment p1/31-36 should be corrected as follow:
Oligonucleotide therapeutics is a general term for state-of-art molecular-target agents that employ chemically synthesized oligonucleotides of single stranded DNA or RNA backbone with potential specificity, and they can inhibit gene expression or impede protein function by binding to a specific sequence of a target gene or protein [1].
Remove the sentence: In contrast to gene based treatments that induce gene expression from a specific DNA sequence and yield a protein with some biofunction, a therapeutic nucleic acid, acts directly in a living body.
Thank you for the comments. I revised that according to your advice.
3. P2/15: correct the sentence as following: “focusing on the possibility of oligonucleotides to work as a potential diagnostic or therapeutic agents for pancreatic cancer”
Thank you for the comments. I revised that according to your advice.
4. Correct the sentence p2/36-38: “As shown in Figure 1, decoys work upstream of the expression process targeting DNA coding transcription factors; subsequently, antisense DNA, siRNA and miRNA act on the level of target mRNA; and finally, aptamers directly inhibit activity of proteins involved in disease development.
Thank you for the comments. I revised that according to your advice.

Round 3
Reviewer 4 Report
Now the manuscript is suitable for publication. However, minor corrections are needed, as listed below:
P1/l.36: Put coma instead of hyphenation in “(ASOs)- small interfering”
P3/l.3 replace: “Figure 1. Standard schematic of oligonucleotide activities in pathogenesis disease progression.” For : “Therapeutic oligonucleotides act on different stages of pathological gene expression.”
P7/l.16 it cannot be “2'-fluoropyrimidine-modified RNA aptamer” – correct possibly to: “2’-fluoro-uridine modified RNA aptamer”
P9/l.5 Again you use the disease progression instead on inhibition of gene expression at the upstream step. “Decoys, which form a double-stranded DNA structure, inhibit DNA transcription at the initial 6 phase of disease progression.” Please, correct to “Decoys, which form a double-stranded DNA structure, inhibit DNA transcription by blocking activity of the dsDNA-binding transcription factors.”
Author Response
Now the manuscript is suitable for publication. However, minor corrections are needed, as listed below:
P1/l.36: Put coma instead of hyphenation in “(ASOs)- small interfering”
P3/l.3 replace: “Figure 1. Standard schematic of oligonucleotide activities in pathogenesis disease progression.” For : “Therapeutic oligonucleotides act on different stages of pathological gene expression.”
P7/l.16 it cannot be “2'-fluoropyrimidine-modified RNA aptamer” – correct possibly to: “2’-fluoro-uridine modified RNA aptamer”
P9/l.5 Again you use the disease progression instead on inhibition of gene expression at the upstream step. “Decoys, which form a double-stranded DNA structure, inhibit DNA transcription at the initial 6 phase of disease progression.” Please, correct to “Decoys, which form a double-stranded DNA structure, inhibit DNA transcription by blocking activity of the dsDNA-binding transcription factors.”
Thank you for the comments. I revised all relevant parts according to your advice.
